# Early Neonatal Cardiac Phenotype in Hurler Syndrome: Case Report and Literature Review

**DOI:** 10.3390/genes13081293

**Published:** 2022-07-22

**Authors:** Nishitha R. Pillai, Alia Ahmed, Todd Vanyo, Chester B. Whitley

**Affiliations:** Advanced Therapies Program, M Health, University of Minnesota, Minneapolis, MN 55454, USA; ahmed306@umn.edu (A.A.); vanyo005@umn.edu (T.V.); whitley@umn.edu (C.B.W.)

**Keywords:** Hurler syndrome, MPS IH, newborn screening, enzyme replacement therapy, hematopoietic stem cell transplant, cardiomyopathy

## Abstract

Mucopolysaccharidosis type I (MPS I) is a rare inherited lysosomal disorder caused by deficiency of the α-L-iduronidase enzyme, resulting in the progressive accumulation of glycosaminoglycans (GAGs), which interfere with the normal function of multiple tissues and organs. The clinical phenotype includes characteristic facial features, hepatosplenomegaly, dysostosis multiplex, umbilical and inguinal hernias, progressive cognitive deficits with corresponding hydrocephalus, and neuropathology. Untreated children do not survive into the second decade. The common cardiac phenotype seen in MPS I and other MPS types includes valve thickening and dysfunction, conduction abnormalities, coronary artery disease, and cardiomyopathy—usually seen later in the disease course. A 15-month-old ex-35-weeker who presented with cardiomyopathy and left ventricular failure at the age of three weeks is presented here. Early evaluation and diagnosis with the help of newborn screening (NBS), followed by treatment with enzyme replacement therapy (ERT) and hematopoietic stem cell transplantation (HSCT), resulted in improvement of his cardiopulmonary status. In MPS I, an early cardiac phenotype is uncommon. Based on the evidence from the literature review for early neonatal cardiac phenotype, we propose that all infants with abnormal newborn screening for MPS I should receive cardiac screening with echocardiogram and NT-proB-type natriuretic peptide (BNP) during the initial evaluation.

## 1. Introduction

Mucopolysaccharidosis type I (MPS I) is a rare inherited lysosomal disorder caused by the deficiency or absence of the α-L-iduronidase enzyme, which contributes to the degradation of glycosaminoglycans (GAGs), and is characterized by the progressive systemic deposition of GAGs in organs and tissues [1,2]. Three major clinical subgroups are usually manifested in mucopolysaccharidosis type I: Hurler (MPS IH, severe), Hurler–Scheie (MPS IH/S, intermediate), and Scheie (MPS IS, mild). Hurler syndrome, which is the most severe form MPS I, has an incidence of about 1 in 100,000 live births [3]. In 2016, mucopolysaccharidosis type I (MPS I) was added to the Recommended Uniform Screening Panel (RUSP). Minnesota started screening for MPS I on 1 August 2017, and as of January 2022, there are 30 states performing newborn screening (NBS) for MPS type I [4]. The list of states screening for MPS I is expanding on an ongoing basis.

The deposition of GAG (dermatan sulfate and heparan sulfate) causes enlargement and thickening of various organs, including the heart, liver, spleen, connective tissues, muscles, and joints, along with corneal clouding, and can also accumulate in the central nervous system, causing various levels of cognitive and functional impairment. Cardiac involvement is a very common and well-documented feature in Hurler syndrome, but usually presents in late infancy. Without treatment, patients experience progressive multisystem manifestations, and usually die within the first decade of life [5].

The current treatment modalities for MPS IH include enzyme replacement therapy (ERT) and hematopoietic stem cell transplantation (HSCT). Laronidase, a recombinant α–L-iduronidase, received Food and Drug Administration (FDA) approval in the United States in 2003, and has since been widely used [6]. ERT has been very effective in reducing visceromegaly and reducing urinary GAG excretion. The inability to cross the blood–brain barrier—resulting in poor neurological outcomes—the possibility of antibody formation reducing the efficacy of the treatment, and the incomplete correction of skeletal, cardiac, and ophthalmologic phenotypes limit its use in severe forms of the disease [7]. HSCT is now the standard of care for individuals with MPS IH. The donor stem cells cross the blood–brain barrier, providing the enzyme that is deficient, and resulting in stabilization of the neurological phenotype [8].

A case of Hurler syndrome in a male ex-35-weeker diagnosed by NBS, who presented with left heart failure at the age of three weeks, is described here. Although rare, the literature review reveals that a few infants have been found to have a cardiac phenotype with minimal systemic findings. Hence, it is recommended to pursue a complete cardiac assessment at the time of initial evaluation after an abnormal NBS for Hurler syndrome.

## 2. Case Report

Our patient was a 15-month-old boy, who initially presented with cardiomyopathy at the age of three weeks. He was born at 35 6/7 weeks gestation, weighing 3.929 kg (87th percentile; z score: 1.13). The rest of the growth parameters included a length of 54 cm (93rd percentile; z score: 1.53) and head circumference of 35.6 cm (44th percentile; z score: −0.13). The delivery was complicated by breech presentation, necessitating cesarean section, and the patient was large for his gestational age. He had an abnormal NBS that was concerning for a high dermatan sulfate level of 2400 nmol/L (reference range: <200) and heparan sulfate level of 608 nmol/L (reference range: <96) at the age of 5 days. He had an uncomplicated postnatal period, and passed congenital heart disease screening, with 99% oxygen saturation in the right hand and 98% in the right foot. He passed the newborn hearing screening in the right ear, while he failed on the left side.

Family history was significant, as the patient’s older sister presented with coarse dysmorphic features, development delays, and sensorineural hearing loss at the age of 9 months, leading to the diagnosis of MPS IH. She had a successful HSCT at the age of 14 months. With the history of Hurler syndrome in this older sibling, our patient underwent a molecular analysis (del/dup and sequencing) of α-L-iduronidase (*IDUA*) gene, revealing a homozygous pathogenic variant c.208C > T (p.Gln70*/Q70X). He was started on ERT with weekly laronidase infusion at 27 days of life at 0.5 mg/kg/week. His initial cardiac evaluation at the age of three weeks showed a moderately depressed ejection fraction (EF) of 28%, with moderately depressed left ventricular systolic function and mild-to-moderate left ventricular enlargement. A patent foramen ovale with a 6 mm Hg gradient across it suggested elevated left atrial pressure and mild mitral regurgitation. Physical examination was completely normal except for mild tachypnea and a small umbilical hernia. Ophthalmological evaluation around the same age showed mild corneal clouding. Audiological examination detected hearing loss that was largely sensorineural.

Around 6 weeks of age, he was admitted to the hospital for worsening of the EF (27%) and left ventricular function, resulting in the initiation of milrinone infusion despite weekly ERT. He was otherwise in his normal state of health. He had received three doses of ERT before admission. Intravenous milrinone therapy at 0.75 mcg/kg/min, along with the ERT, improved EF to 50% during that admission. The patient was discharged home on 0.5 mcg/kg/min of milrinone with an EF of 42%. His N-terminal-pro BNP was elevated at 9382 pg/mL (reference range: 0–1000), and troponin was slightly above normal at 0.065 ug/L (reference range: 0–0.045). He was evaluated for bone marrow transplantation, but it was somewhat delayed due to his refractory cardiac phenotype. A cardiomyopathy panel sent to evaluate for other possible causes of cardiomyopathy returned normal, without any variants in any of the tested genes. A neuropsychological evaluation conducted at the age of 4 months before HSCT showed his development to be appropriate for his age. A skeletal survey conducted at the age of 4 months was normal except for a J-shaped sella turcica and mild beaking of the inferior lumbar vertebrae. Physical examination was not suggestive of any hepatosplenomegaly.

The patient had an unrelated 7/8 human leukocyte antigen (HLA)-matched umbilical cord blood transplant at the age of 4 months. The post-HSCT course was uneventful, with a hospitalization that lasted 24 days. He was weaned off the milrinone infusion and had an EF of 62% at the time of discharge. He had 100% engrafting in the myeloid compartment on day 21 post-HSCT, and remained fully engrafted during the recent clinic visit, one year post-transplant. He continues to be on ERT with laronidase at 0.5 mg/kg every week at 15 months of age. His most recent neuropsychology evaluation at the age of 11 months (6 months post-HSCT) showed continued acquisition of developmental skills across all domains, without any developmental regression. Six months post-HSCT, α-iduronidase activity in the leukocytes was normal at 24.78 nmol/h/mg (reference range: 6.0–71.4), and in the plasma it was slightly low at 2.37 nmol/h/mg (reference range: 3.0–50.2), and the patient had negative IgG anti-laronidase antibody status. His plasma GAG showed near-normal concentrations of dermatan sulfate (132 nmol/L; reference range: </=130) and normal heparan sulfate (43 nmol/L; reference range: </=95), while urine GAG showed increased excretion of heparan sulfate (0.84 mg/mmol Cr; reference range: </=0.50) and dermatan sulfate (1.10 mg/mmol Cr; reference range: </=1.00). His N-terminal BNP continues to be normalized at 1 year post-HSCT, at 78 pg/mL (reference range: 0–1000), and with troponin I at <0.015 μg/L (reference range: 0–0.045).

## 3. Methods

Quantitative analysis of total GAGs in urine was performed using a 1,9-dimethylene blue (DMB) colorimetric reaction that was measured by spectrophotometry at a wavelength of 656 nm. GAG measurements were reported relative to the creatinine concentration in the patient’s urine. Quantification of individual glycosaminoglycans, dermatan sulfate, and heparan sulfate was performed using liquid chromatography–tandem mass spectrometry (LC–MS/MS).

The blood specimens were eluted and sonicated. Dermatan sulfate, heparin sulfate (HS), and keratan sulfate were enzymatically digested. The reaction mixture was centrifuged and analyzed by LC–MS/MS. The ratio of the extracted peak area of DS, HS, and KS to the internal standard, as determined by LC–MS/MS, was used to calculate the concentrations of DS, HS, and KS in the sample.

Next-generation sequencing and copy number variation analysis were performed as a part of the evaluation of the genes in the cardiomyopathy panel. 

## 4. Discussion

The onset of clinical symptoms in Hurler syndrome depends on the severity of the phenotype. Onset of symptoms in infancy is suggestive of MPS I Hurler syndrome, while the onset might be delayed up to the second decade in attenuated form or the MPS I Scheie disease [9]. The initiation of newborn screening, leading to the early diagnosis of MPS I, provides a significant insight into the genotype–phenotype association in this disease. Data from the international MPS I registry show that variants predicted to severely disrupt gene transcription or translation (e.g., nonsense variants, splice site disruption, or initiator codon variants and frameshifts), when present in a homozygous or compound heterozygous state, result in a severe phenotype [10]. Similarly, missense variants, when in a homozygous state or in trans with a variant predicted to be severely disruptive to gene transcription or translation, lead to an attenuated phenotype [10]. As the GAGs accumulate, manifestations such as corneal clouding, hearing loss, and respiratory, cardiac, and neurological issues occur. 

The genotype seen in our patient, Q70X, is one of the common *IDUA* variants seen among Caucasians—especially from Scandinavia and Russia—and has been associated with a severe clinical phenotype [11,12,13]. This variant results in an unstable and truncated IDUA protein that is incapable of binding to the substrate [14]. Phenotypical signs and symptoms associated with this specific genotype have been reported as early as 2 months of life [15].

Cardiac manifestations in MPS I typically occur around 8–10 months of age [16]. The most common cardiac phenotype seen in MPS—especially MPS IH—includes cardiac valve thickening and dysfunction, conduction abnormalities, and coronary artery disease [17]. The presence of dilated as well as hypertrophic cardiomyopathy has also been well documented in MPS IH [18,19,20,21]. It is believed that accumulation of GAGs within myocardiocytes and the myointima of coronary arteries results in cardiomyopathy in these children [22]. It was initially believed that cardiac manifestations—especially ventricular hypertrophy present in these patients—occur later in the disease progression, after other physical and developmental abnormalities have already been identified. Recently, more neonates have been found to have early cardiac manifestations, leading to the diagnosis of MPS I (Table 1).

The very first report of cardiomyopathy in infancy was published by Donaldson et al., where he described five children who presented with cardiomyopathy within their first year of life [19]. The earliest cardiac phenotype reported initially in Hurler syndrome was at 10 weeks of age, where the patient presented with features of severe heart failure and was treated with ERT and matched unrelated donor HSCT [20]. Wiseman et al., in 2013, reported six infants with MPS IH who presented with cardiomyopathy within their first 4 months of life, of whom two patients presented at 1 week and 2.5 weeks of age, respectively [21]. Recently, Miselli et al. described a 13-day-old neonate who presented with feeding difficulties and respiratory distress, which led to the diagnosis of dilated cardiomyopathy and, eventually, MPS I [23]. 

The natural history of MPS has shown that cardiac abnormalities in these individuals worsen with age. It has been demonstrated that initiating ERT in these patients can help to stabilize and even reverse cardiomyopathy [24,25,26]. As a result, early discovery and initiation of treatment are crucial for improved outcomes, which can be accomplished by neonatal screening. Recommended follow-up guidelines for an abnormal NBS for MPS I have already been published [27]. Once the diagnosis is confirmed, the recommended follow-up surveillance includes serial skeletal radiographs, echocardiography, neurocognitive evaluation, ophthalmologic examination, and consideration of brain magnetic resonance imaging (MRI) for early identification [27,28]. While adhering to the recommended surveillance by Clarke et al., it is recommended that an echocardiogram, pro-BNP, and troponin I readings be obtained at the time of the initial evaluation, in addition to complete physical examination, molecular testing for IDUA variants, urine GAG, α-L-iduronidase activity, and a comprehensive metabolic panel, due to the increased prevalence of isolated early cardiac phenotypes (Table 2).

HSCT has been proposed as an effective treatment for Hurler syndrome, and clinical trials have subsequently shown that it reduces hepatosplenomegaly, urinary GAG, and intracranial pressure (hydrocephalous), along with stabilization of cognitive decline, increasing survival [29,30,31]. HSCT has since become the standard of care wherever transplantation is available. HSCT has also been shown to stabilize neuropsychological function in children with MPS IH [32,33,34]. Delivery of enzymes by the engrafted donor cells has a favorable effect on the visceral manifestations, including the different cardiac features seen in this disease [35]. 

It is also important to consider the limitations of HSCT. Corneal clouding and skeletal complications may continue to progress despite HSCT [36]. Similarly, HSCT may not completely correct valvular heart disease or reverse preexisting cognitive defects [37,38].

The combination therapy where ERT is initiated before HSCT can provide a preferable environment for donor engraftment by reducing the GAG deposition in the marrow and limiting the pulmonary complications after HSCT, thereby reducing morbidity and mortality [39,40]. A combination of extended pre-transplant ERT with HSCT has been shown to improve cardiac outcomes in infants with severe cardiomyopathy [21]. Polgreen et al. recently evaluated the impact of ERT treatment in a post-HSCT population [41]. Improvement in linear growth in younger individuals, along with an overall quality of life denoted by an improvement in the six-minute walk test, has been reported as an advantage of post-HSCT enzyme augmentation—especially when the patients did not develop anti-laronidase antibodies [41]. Some of the urinary GAGs originate from the luminal urothelial surface, and the rest from the renal clearance of circulatory GAGs [42]. Since previous studies have shown that patients on ERT had normalization of serum GAGs before urine, it was proposed that the GAG pools may not be equivalent [43]. As a result, plasma/serum may be less useful than urine in monitoring the disease. 

With the increasing prevalence of genetic testing, a higher incidence of dual diagnosis has been detected [44,45]. The possibility of an additional diagnosis for cardiomyopathy was evaluated in our patient using an extensive dilated cardiomyopathy panel (42 genes: *ABCC9*, *ACTC1*, *ACTN2*, *BAG3*, *CASZ1*, *CRYAB*, *CSRP3*, *DES*, *DMD*, *DNAJC19*, *DSG2*, *DSP*, *EYA4*, *FKTN*, *GATAD1*, *LAMA4*, *LDB3*, *LMNA*, *MYBPC3*, *MYH6*, *MYH7*, *MYPN*, *NEXN*, *NRAP*, *PLN*, *PPCS*, *PRDM16*, *PSEN1*, *PSEN2*, *RAF1*, *RBM20*, *SCN5A*, *SGCD*, *TCAP*, *TMPO*, *TNNC1*, *TNNI3*, *TNNI3K*, *TNNT2*, *TPM1*, *TTN*, and *VCL*), which did not show any variants. 

## 5. Conclusions

The significant improvement in cardiopulmonary function in our patient from early diagnosis and treatment with ERT and HSCT substantiates the importance of newborn screening in this group of disorders. It is strongly recommended that healthcare providers—including geneticists—consider the possibility of MPS in early neonatal cardiomyopathy, even in the absence of other associated clinical features. Due to the increased incidence of cardiac phenotype in the early neonatal period, it is important that the initial evaluation after an abnormal NBS should include echocardiogram and N-terminal Pro-BNP, irrespective of the symptoms. Regular monitoring of these patients by a cardiologist familiar with the cardiac manifestations of the MPS diseases is recommended once the diagnosis is established.

## Figures and Tables

**Table 1 genes-13-01293-t001:** Early cardiac phenotype in Hurler syndrome.

	Number of Patients	S/S at Birth	Age of Onset of Cardiac Phenotype	Cardiac Phenotype	Biochemical Abnormality	Abnormal NBS	Genotype	Treatment
Donaldson et al. [19].	P1	Hip dysplasia	5 months	Acute HF	NA	NA	NA	NA
	P2	Postural deformities	3 months	Acute HF, LVH	High urine GAG	NA	NA	NA
	P3	Scoliosis, hip dysplasia	11 months	Sudden death	NA	NA	NA	NA
	P4	Inguinal hernia	5 months	Acute HF, LVH	High urine GAG, low enzyme	NA	NA	NA
	P5	NA	3 months	Acute HF, LVH	NA	NA	NA	NA
Hirth et al. [20].	P1	Healthy	10 weeks	Dilated CM and severe HF	High urine GAG	NA	NA	ERT started at 10 weeks and HSCT at 11 months
Wiseman et al. [21].	P1	NA	3 months	Dilated CM with LVD	High urine GAG, low enzyme	NA	Homozygous c.705insTGCTC	ERT for 4 months, HSCT at 14 months
	P2	NA	1 week	Dilated CM with LVD	High urine GAG, low enzyme	NA	Homozygous R628X	ERT for 2 months, HSCT at 17 months
	P2	NA	4 months	Dilated CM with LVD	High urine GAG, low enzyme	NA	Homozygous W402X	ERT for 9 months, HSCT at 13 months
	P4	NA	3 months	Dilated CM with LVD	High urine GAG, low enzyme	NA	Homozygous R628X	ERT for 5 months, HSCT at 12 months
	P5	NA	18 days	Dilated CM with LVD	High urine GAG, low enzyme	NA	Homozygous R628X	ERT for 8 months, HSCT at 9 months
	P6	NA	3 months	Dilated CM with LVD	High urine GAG, low enzyme	NA	Q70X/W402X	ERT for 4 months, HSCT at 8 months
Miselli et al. [23].	P1	Respiratory distress, feeding difficulties	13 days	CM with LVD	High urine GAG, low enzyme	NA	Homozygous c.46_587del12	ERT for 2 months, HSCT at 2 months
Current patient	P1	Mild tachypnea, umbilical hernia	3 weeks	CM with LVD	High urine GAG, low enzyme	Yes	Homozygous p.Q70X	ERT started at 27 days and HSCT at 4 months

**Table 2 genes-13-01293-t002:** Recommended testing following an abnormal newborn screening for MPS I.

Recommended Evaluation after an Abnormal Newborn Screening	
Comprehensive metabolic panel	√
Electrocardiogram	√
Echocardiogram	√
NT-proB-type natriuretic peptide (Pro-BNP)	√
Enzyme analysis	√
Urine glycosaminoglycans	√
Molecular analysis of IDUA (if not included in the NBS)	√

## Data Availability

Not applicable.

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
