# Peer review of "Early Neonatal Cardiac Phenotype in Hurler Syndrome: Case Report and Literature Review"

_genes, 2022, doi:10.3390/genes13081293_

Round 1

Reviewer 1 Report

Manuscript ID: genes-1804481

Type of manuscript: Case Report

Title: Early Neonatal Cardiac Phenotype in Hurler Syndrome: Case Report and Literature Review

This case report described a three-week-old ex-35 weeker who presented with cardiomyopathy and left ventricular failure at the age of three weeks. Early evaluation and diagnosis with the help of newborn screening (NBS) followed by treatment with enzyme replacement therapy (ERT) and hematopoietic stem cell transplantation (HSCT) resulted in improvement of his cardiopulmonary status. Based on the evidence from the literature review for early neonatal cardiac phenotype, the authors proposed that all infants with abnormal newborn screen for MPS I should have cardiac screening with echocardiogram and NT-proB-type Natriuretic Peptide (BNP) during the initial evaluation.

This case report is important and interesting, however, there are some points needed to be clarified or revised.

(1)   Line 39: “heparin” should be changed to “heparan”.

(2)   Line 53, Please add the data of birth height and birth head circumference of this patient in the manuscript.

(3)   Line 55: What is the data of IDUA enzyme activity? Please add this data in the manuscript.

(4)   Line 55: What is the screening method of NBS for MPS I in this study? The author described that “He had an abnormal NBS that was concerning for high dermatan sulfate of 2400 nmol/L (reference range: < 200) and heparan sulfate of 608 nmol/L (reference range: < 96) at the age of 5 days”. Do the authors check dermatan sulfate and heparan sulfate levels instead of IDUA enzyme activity in the dried blood spot of NBS?

(5)   Line 59: …in the right “had” and…? Please check the word “had”.

(6)   Page 68: At the age of three weeks, what is the imaging results of bone X-ray and abdominal ultrasonography for this patient? Did the patient have bone deformities or hepatosplenomegaly at that time?

(7)   Line 92: The author described that “He had 100% engrafting in the myeloid compartment on day 21 post HSCT and remains fully engrafted during the recent clinic visit, one-year post transplant. He continues to be on ERT with laronidase at 0.5 mg/kg every week”. Since HSCT seems successful for this patient, when will the patient plan to stop ERT?

(8)   Line 95: What is the data of “N terminal-pro BNP” and “troponin” after ERT and HSCT for this patient? Please add these data in the manuscript.

(9)   Line 97: In addition to “total antibodies (TAbs) against laronidase”, did the authors also check “Neutralizing antibodies (NAbs) against laronidase” for this patient? If the answer is yes, please also add the data of “Neutralizing antibodies (NAbs) against laronidase” in the manuscript.

Author Response

Reviewer 1:

Abstract

  1. 12-14: The common cardiac phenotype (valve thickening and dysfunction, conduction abnormalities, coronary artery disease and cardiomyopathy) refers to all types of MPS types, not just to “MPS IH”.

Thank you for pointing this out. We have corrected the sentence. Please see page 2, line 7:

The common cardiac phenotype seen in MPS I and other MPS types includes valve thickening and dysfunction, conduction abnormalities, coronary artery disease and cardiomyopathy.

  1. 14, and ln. 18-19: the authors state that cardiac pathology in MPS IH is “usually seen later in the disease course”. This does not correspond to the data reported in the literature, showing that cardiac pathology generally develops earlier in life for individuals with more rapidly progressing types of MPS (e.g. Hurler syndrome) and may be delayed in slowly progressing forms (e.g. Hurler-Scheie and Scheie syndromes). In addition, it is in stark contrast to the data reported in Table 1 by the authors.

We have changed this accordingly. Please see page 2, line 7. The manuscript now reads:

The common cardiac phenotype seen in MPS I and other MPS types includes valve thickening and dysfunction, conduction abnormalities, coronary artery disease and cardiomyopathy.

Introduction

  1. 43: The authors again state that cardiac involvement in Hurler syndrome “usually presents in late infancy”. However, moderate to severe narrowing of the coronary arteries can occur within the first year of life and complete occlusion of the coronary arteries has been reported within the first 5 years of life of Hurler patients. In addition, heart failure has been seen within the first few weeks of life in Hurler syndrome.

Please, see the references cited in Table 1 and others. For example,

Kiely, B.T., Kohler, J.L., Coletti, H.Y. et al. Early disease progression of Hurler syndrome. Orphanet J Rare Dis 12, 32 (2017). https://doi.org/10.1186/s13023-017-0583-7;

Miselli, F., Brambilla, A., Calabri, G. B., Favilli, S., Sanvito, M. C., Ragni, L., Torcetta, F., Rossi, K., Donati, M. A., & Procopio, E. (2021). Neonatal heart failure and noncompaction/dilated cardiomyopathy from mucopolysaccharidosis. First description in literature. Molecular genetics and metabolism reports, 26, 100714. https://doi.org/10.1016/j.ymgmr.2021.100714;

 Hampe, C. S., Eisengart, J. B., Lund, T. C., Orchard, P. J., Swietlicka, M., Wesley, J., & McIvor, R. S. (2020). Mucopolysaccharidosis Type I: A Review of the Natural History and Molecular Pathology. Cells, 9(8), 1838. https://doi.org/10.3390/cells9081838

We thank you for providing the additional references. What the authors meant was that as per the literature review, cardiac manifestations are usually seen late in infancy. However, as we learn more about MPS IH, isolated cardiac manifestations in the absence of other systemic symptoms could be seen even in the early neonatal period.

As per the references the reviewer has provided, (Kiley et al), the median age at the initial detection of any cardiac abnormality was 9.5 months and as per Hampe et al, ventricular dysfunctions usually do not present until 6 months of age, followed by reduction of contractile function and ejection function by 10 months of age. Hence, we thought, it would be appropriate to say that cardiac phenotype presents in late infancy.        

  1. The authors should introduce in this Section some brief information on the efficacy of ERT and HSCT therapeutic approaches used for MPS I treatment, as the patient was subjected to these treatments (see Case report).

This has been added to the manuscript. Please see page 3, paragraph 3, line, that now reads:

The current treatment modalities for MPS IH include enzyme replacement therapy (ERT) and hematopoetic stem cell transplantation (HSCT). Laronidase, a recombinant alpha‐L‐iduronidase received Food and Drugs Administration (FDA) approval in United States in 2003 and has been since widely used (Jameson et al 2019). ERT has been very effective in reducing visceromegaly and reducing urinary GAG excretion. The inability to cross the blood brain barrier resulting in a poor neurological outcome, possibility of antibody formation reducing the efficacy of the treatment, and incomplete correction of skeletal, cardiac and ophthalmologic phenotypes limits its use in severe form of the disease (Parini et al 2017). HSCT is now the standard of care for individuals with MPS IH. The donor stem cells cross the blood brain barrier providing the enzyme that is deficient resulting in stabilization of the neurological phenotype (Taylor et al 2019).

Above mentioned 3 references are also added in the manuscript under reference.

Case report

  1. How long the ERT treatment lasted? (ln. 66-67).

Our patient continues to be on ERT at the age of 15 months. These individuals at our institution continue to be on ERT post HSCT for a minimum of at least 2 years.

Please see page 6, line 6, that now reads:

He continues to be on ERT at 15 months with laronidase at 0.5 mg/kg every week.

  1. How GAG content was measured in the plasma and in the urines of the patients? -

This has been included in the manuscript under the methods section. Please see page 6, paragraph 2 under methods:

 In urine, Quantitative analysis of total glycosaminoglycans (GAGs) is performed using a 1,9-dimethylene blue (DMB) colorimetric reaction that is measured by spectrophotometry at a wavelength of 656 nm. GAG measurements are reported relative to the creatinine concentration in the patient's urine. Quantification of individual glycosaminoglycans, dermatan sulfate (uDS) and heparan sulfate (uHS), is performed using liquid chromatography-tandem mass spectrometry (LC-MS/MS).

The blood spot specimens are eluted and sonicated. Dermatan sulfate (DS), heparin sulfate (HS), and keratan sulfate (KS) are enzymatically digested. The reaction mixture is centrifuged and analyzed by LC-MS/MS. The ratio of the extracted peak area of DS, HS, and KS to internal standard as determined by LC-MS/MS is used to calculate the concentration of DS, HS, and KS in the sample.

Discussion

  1. The authors list the benefits of HSCT treatment for Hurler syndrome (lns. 166-174). However, they should also discuss the limits of such an approach. Indeed, it is known that, in spite of successful engraftment, the cardiac valves do not receive the beneficial effects of transplantation. The mitral and aortic valves continue to thicken and may develop leaking and/or stenosis. Because of valve abnormalities, virtually children require prophylaxis with antibiotics (SBE prophylaxis) at the time of dental or other contaminated procedures.

The limitations of HSCT have been added to the manuscript. Please see page 9, paragraph 2:

It is also important to consider the limitations of HSCT. Corneal clouding and skeletal complications may continue to progress despite HSCT (Kurtzberg 2015). Similarly, HSCT may not completely correct valvular heart disease or reverse preexisting cognitive defects (Kubaski et al 2020; Guffon et al 2021).

Above mentioned 3 references are also added in the manuscript under reference.

  1. In addition to the recommended evaluation following an abnormal newborn screen for MPS I reported in Table 2, the authors should also outline that a regular follow-up by cardiac ultrasound, electrocardiogram, and Holter monitoring by a cardiologist familiar with the cardiac manifestations of the MPS diseases is important for individuals with MPS I.

Thank you for emphasizing this. We have added this important information in the conclusion. Please see page 11, paragraph 1, line 8:

Regular monitoring of these patients by a cardiologist familiar with the cardiac manifestations of the MPS diseases is recommended once the diagnosis is established.

  1. What methodology did the authors use to evaluate the dilated cardiomyopathy gene panel? (ln. 187-193).

Next generation sequencing and copy number variation analysis were done for evaluation of the genes mentioned in the manuscript.

Genomic DNA is extracted from the sample, and sequencing libraries are prepared according to standard Agilent protocols using a custom-designed Agilent Sureselect XT kit for enrichment of targeted genes. The enriched DNA libraries are sequenced on an Illumina Novaseq or Nextseq instrument. Raw sequencing reads are mapped to the reference genome using BWA. Raw alignment files are realigned in the neighborhood of indels and recalibrated for base quality accuracy using the Genome Analysis Tool Kit (GATK) version 3.8. Point mutation and indel calls in exons and adjoining intronic regions are made using a combination of the GATK haplotype caller and Samtools mpileup. Variants are interpreted according to guidance issued by the American College of Medical Genetics.

For copy number variation (CNV) analysis, raw sequencing reads are mapped to the reference human genome (HG19) using both BWA and Bowtie algorithms. CNV ratios are computed by comparing the average coverage in the sample across 60 base pair windows. Average coverage is compared to control loci both within the sample and within a gender-matched control sample from the same run. The BWA/Bowtie coverage ratio is calculated for each window in order to identify regions where the presence of homologous sequences precludes accurate CNV calls. Heterozygous deletion calls are made when 3 consecutive windows have a CNV ratio of 0.3-0.7; homozygous/hemizygous deletion calls are made when 3 consecutive windows have a CNV ratio less than 0.3; and duplication calls are made when 5 consecutive windows have a CNV ratio greater than 1.3. All CNV calls are confirmed with a supplementary qPCR assay.

  1. There is no discussion on the levels of GAGs in the plasma and in the urines of the patient as measured six months after HSCT. Why the levels of heparan sulfate and dermatan sulfate in the plasma resulted to be in the normal ranges, while in the urines were still increased?

We have included the proposed etiology in the manuscript. Please see page 10, paragraph 3, line 10, that now reads:

Some of the urinary GAGs originate from the luminal urothelial surface and the rest from the renal clearance of circulatory GAGs (Hurst 1994). Since previous studies showed that patients on ERT had normalization of serum GAGs before urine, it was proposed that the GAG pools may not be equivalent (Zhang et al 2020). As a result, plasma/serum may be less useful than urine in monitoring the disease.

  1. The Sections “Author Contributions”, “Funding”, “Institutional Review Board Statement”, “Data Availability Statement” must be completed.

These sections have been completed and is included in the manuscript now. Please see page 12

AUTHOR CONTRIBUTIONS: NRP and AA conceptualized and drafted the original manuscript. TV and CW reviewed and revised the manuscript. All authors participated in critical review of the manuscript for important intellectual content. All authors approved the final manuscript as submitted and agree to be accountable for all aspects of the work.

FUNDING/SUPPORT: No funding was secured for this study.

INSTITUTIONAL REVIEW BOARD STATEMENT: An IRB approval for case reports of a single patient is not required at the University of Minnesota.

DATA AVAILABILITY STATEMENT:  Data sharing not applicable to this article as no datasets were generated or analysed during the current study

  1. 6, ln. 207: In “Supplemetary Materials”, “Figure S1: title; Table S1: title; Video S1: title” are cited but are not mentioned in the text.

No “Supplementary Materials” are attached. We do not have figure S1, table S1, video S1 in the manuscript. This was automatically uploaded by the journal website. We have requested them to rectify this mistake.

Minor points

  1. Define the abbreviation NBS (newborn screening) not in the Abstract (ln.16) but in the Introduction at ln. 36.

This has been corrected. Please see page 3, paragraph 1, line 6 which now reads:

Minnesota started screening for MPS I on August 1, 2017, and as of January 2022, there are 30 states

are doing newborn screening (NBS) for MPS type I (Ojodu et al 2018). The list of states screening for

MPS I is expanding on an ongoing basis.

  1. 1, ln. 9: add “with” after”interfering”.

This has been corrected. Please see page 2, line 1, which now reads:

Mucopolysaccharidosis type I (MPS I) is a rare inherited lysosomal disorder caused by deficiency of α-L-

iduronidase enzyme resulting in the progressive accumulation of glycosaminoglycans (GAG) interfering

with the normal function of multiple tissues and organs.

  1. 1, ln. 6: delete comma after “untreated”.

This has been corrected. Please see page 2, line 6, which now reads:

Untreated children do not survive into the second decade.

  1. 2, ln. 13: define the abbreviation “HSCT”.

This has been corrected. Please see page 2, line 13, which now reads:

Early evaluation and diagnosis with the help of newborn screening (NBS ) followed by treatment with

enzyme replacement therapy (ERT) and hematopoietic stem cell transplantation (HSCT) resulted in

improvement of his cardiopulmonary status.

  1. 2, ln. 65: define the abbreviation “IDUA”, which now reads”:

This has been corrected. Please see page 5, line 1.

With the history of Hurler syndrome in this older sibling, our patient had a molecular analysis (del/dup

and sequencing) of α-L-iduronidase (IDUA) gene which revealed a homozygous pathogenic variant

c.208C>T (p.Gln70*/ Q70X).

  1. 2, ln. 66: define the abbreviation “ERT”

This has been corrected. Please see page 3, paragraph 3, line 1, which now reads:

He was started on enzyme replacement therapy (ERT) with weekly laronidase infusion at 27 days of life

at 0.5mg/kg/week.

  1. 7, ln. 122: spell “have”

This has been corrected. Please see page 7, paragraph 3, line 5, which now reads:

Phenotypical signs and symptoms associated with this specific genotype have been reported as early

as 2 months of life (Vazna et al 2009)

  1. 8, ln. 124: spell “includes”, which now reads:

This has been corrected. Please see page 8, paragraph 1, line 2, which now reads:

The most common cardiac phenotype seen in MPS especially MPS IH includes cardiac valve thickening

and dysfunction, conduction abnormalities and coronary artery disease (Braunlin et al 2011).

  1. 8, paragraph 1, line 8: spell “occur”.

This has been corrected. Please see page 7, line 171, which now reads:

It was initially believed that cardiac manifestations, especially ventricular hypertrophy present in these

patients, occur later in the disease progression, after other physical and developmental abnormalities

have already been identified.

  1. 9, ln. 162: use the abbreviation “GAG” for “glycosaminoglycans”.

This has been corrected. Please see page 9, paragraph 1, line 10, which now reads:

While agreeing to the recommended surveillance by Clarke et al it is recommended that an

echocardiogram, pro BNP, and troponin I be obtained at the time of the initial evaluation in addition to

complete physical examination, molecular testing for IDUA variants, urine GAG, alpha-L-iduronidase

activity, and comprehensive metabolic panel due to the increased prevalence of isolated early cardiac

phenotype.

  1. 11, ln. 198: use the abbreviation “MPS” for “mucopolysaccharidoses”. –

This has been corrected. Please see page 11, line 3, which now reads:

It is strongly recommended that health care providers including geneticists consider the possibility of

MPS in early neonatal cardiomyopathy even in the absence of other associated clinical features.

  1. 11, ln. 238: who does “his” refer to?

This has been corrected. Please see page 12, under the section Acknowledgements:

The authors would like to thank the patients, their family and all physicians involved in our patient’s care.

  1. 6, ln. 203-205: abolish the heading “Patents” and the entire sentence “This is not mandatory…..”. –

It is unclear regarding what the reviewer is mentioning in this comment since such a sentence is not included in the manuscript. We request the journal to look into the possibility of autopopulating this sentence.

Reviewer 2 Report

In this manuscript, the authors describe a case report of an early neonatal cardiac phenotype in a child affected by Hurler syndrome. The patient at 27 days of life started to receive weekly infusions of laronidase (ERT therapy), and at the age of 4 months was subjected to HSCT which resulted in an improvement of its cardiac status. The authors also provide a limited overview of the literature on the early neonatal cardiac phenotype in Hurler syndrome and suggest that all infants with abnormal newborn screen for MPS I should have cardiac screening with echocardiogram and NT-proB-type Natriuretic Peptide (BNP) during the initial evaluation. 

Although the topic is interesting, there are some issues that need to be addressed.

Abstract

·      Lns. 12-14: The common cardiac phenotype (valve thickening and dysfunction, conduction abnormalities, coronary artery disease and cardiomyopathy) refers to all types of MPS types, not just to “MPS IH”. 

·      Ln. 14, and ln. 18-19: the authors state that cardiac pathology in MPS IH is “usually seen later in the disease course”. This does not correspond to the data reported in the literature, showing that cardiac pathology generally develops earlier in life for individuals with more rapidly progressing types of MPS (e.g. Hurler syndrome) and may be delayed in slowly progressing forms (e.g. Hurler-Scheie and Scheie syndromes). In addition, it is in stark contrast to the data reported in Table 1 by the authors.

Introduction

1.     Ln. 43: The authors again state that cardiac involvement in Hurler syndrome “usually presents in late infancy”. However, moderate to severe narrowing of the coronary arteries can occur within the first year of life and complete occlusion of the coronary arteries has been reported within the first 5 years of life of Hurler patients. In addition, heart failure has been seen within the first few weeks of life in Hurler syndrome.

Please, see the references cited in Table 1 and others. For example, 

·      Kiely, B.T., Kohler, J.L., Coletti, H.Y. et al. Early disease progression of Hurler syndrome. Orphanet J Rare Dis 12, 32 (2017). https://doi.org/10.1186/s13023-017-0583-7

·      Miselli, F., Brambilla, A., Calabri, G. B., Favilli, S., Sanvito, M. C., Ragni, L., Torcetta, F., Rossi, K., Donati, M. A., & Procopio, E. (2021). Neonatal heart failure and noncompaction/dilated cardiomyopathy from mucopolysaccharidosis. First description in literature. Molecular genetics and metabolism reports26, 100714. https://doi.org/10.1016/j.ymgmr.2021.100714;

·      Hampe, C. S., Eisengart, J. B., Lund, T. C., Orchard, P. J., Swietlicka, M., Wesley, J., & McIvor, R. S. (2020). Mucopolysaccharidosis Type I: A Review of the Natural History and Molecular Pathology. Cells9(8), 1838. https://doi.org/10.3390/cells9081838

2.     The authors should introduce in this Section some brief information on the efficacy of ERT and HSCT therapeutic approaches used for MPS I treatment, as the patient was subjected to these treatments (see Case report).

Case report

·      How long the ERT treatment lasted? (ln. 66-67).

·      How GAG content was measured in the plasma and in the urines of the patients?

Discussion

·      The authors list the benefits of HSCT treatment for Hurler syndrome (lns. 166-174). However, they should also discuss the limits of such an approach. Indeed, it is known that, in spite of successful engraftment, the cardiac valves do not receive the beneficial effects of transplantation. The mitral and aortic valves continue to thicken and may develop leaking and/or stenosis. Because of valve abnormalities, virtually children require prophylaxis with antibiotics (SBE prophylaxis) at the time of dental or other contaminated procedures.

·      In addition to the recommended evaluation following an abnormal newborn screen for MPS I reported in Table 2, the authors should also outline that a regular follow-up by cardiac ultrasound, electrocardiogram, and Holter monitoring by a cardiologist familiar with the cardiac manifestations of the MPS diseases is important for individuals with MPS I. 

·      What methodology did the authors use to evaluate the dilated cardiomyopathy gene panel? (ln. 187-193).

·      There is no discussion on the levels of GAGs in the plasma and in the urines of the patient as measured six months after HSCT. Why the levels of heparan sulfate and dermatan sulfate in the plasma resulted to be in the normal ranges, while in the urines were still increased? 

The Sections “Author Contributions”, “Funding”, “Institutional Review Board Statement”, “Data Availability Statement” must be completed. 

Pag. 6, ln. 207: In “Supplemetary Materials”, “Figure S1: title; Table S1: title; Video S1: title” are cited but are not mentioned in the text. 

 Minor points

Define the abbreviation NBS (newborn screening) not in the Abstract (ln.16) but in the Introduction at ln. 36.

Pag.1, ln. 9: add “with” after”interfering”.

Pag.1, ln. 12: delete comma after “untreated”.

Pag. 2, ln. 63: define the abbreviation “HSCT”.

Pag. 2, ln. 65: define the abbreviation “IDUA”.

Pag.2, ln. 66: define the abbreviation “ERT”.

Pag.3, ln. 122: spell “have”.

Pag.3, ln. 124: spell “includes”.

Pag. 3, ln. 131: spell “occur”.

Pag.5, ln. 162: use the abbreviation “GAG” for “glycosaminoglycans”.

Pag. 6, ln. 198: use the abbreviation “MPS” for “mucopolysaccharidoses”.

Pag.6, ln. 238: who does “his” refer to?

Pag.6, ln. 203-205: abolish the heading “Patents” and the entire sentence “This is not mandatory…..”. 

Author Response

  • Line 39: “heparin” should be changed to “heparan”.

We apologize for the mistake. This has been corrected. Please see page 3, paragraph 2, line 1, that now reads:

The deposition of GAG (dermatan sulfate and heparan sulfate) causes enlargement and thickening of various organs like the heart, liver, spleen, connective tissues, muscles, joints, corneal clouding, and can also accumulate in the central nervous system causing various level of cognitive and functional impairment

  • Line 53, Please add the data of birth height and birth head circumference of this patient in the manuscript.

These parameters have been added to the manuscript. Please see page 4, paragraph 3, line 3, that now reads:

The rest of the growth parameters include a length of 54 cm (93rd centile; z score: 1.53) and head circumference of 35.6 cm (44th centile; z score: -0.13).

  • Line 55: What is the data of IDUA enzyme activity? Please add this data in the manuscript.

Unfortunately, we do not have a pre HSCT IDUA enzyme activity in the chart. We do have post HSCT enzyme activity. Please see page 6, line 9, that now reads:

Six months post-HSCT, alpha iduronidase activity in the leukocytes was normal at 24.78 nmol/hr/mg (reference range: 6.0 - 71.4) and in the plasma was slightly low at 2.37 nmol/hr/mg (reference range: 3.0 - 50.2) and had negative IgG anti laronidase antibody status.

  • Line 55: What is the screening method of NBS for MPS I in this study? The author described that “He had an abnormal NBS that was concerning for high dermatan sulfate of 2400 nmol/L (reference range: < 200) and heparan sulfate of 608 nmol/L (reference range: < 96) at the age of 5 days”. Do the authors check dermatan sulfate and heparan sulfate levels instead of IDUA enzyme activity in the dried blood spot of NBS?

The State of Minnesota measures alpha iduronidase activity on the dried blood spot by flow injection analysis tandem mass spectrometry (FIA-MS/MS) as a first tier test. If low, a second-tier test of fractionated glycosaminoglycans is done by liquid chromatography-tandem mass spectrometry (LC-MS/MS) by Mayo Clinic Laboratories. The third tier includes full gene analysis by Mayo Clinic Laboratories. When abnormal, Minnesota Department of Health only reports the outcome of second tier and third tier test (fractionated glycosaminoglycan levels and molecular test results).

  • Line 59: …in the right “had” and…? Please check the word “had”.

We have corrected the sentence. Please see page 4, paragraph 3, line 9, that now reads:

He had an uncomplicated post-natal period and passed congenital heart disease screen with 99% oxygen saturation in the right hand and 98% in the right foot.

  • Page 68: At the age of three weeks, what is the imaging results of bone X-ray and abdominal ultrasonography for this patient? Did the patient have bone deformities or hepatosplenomegaly at that time?

Please see page 5, line paragraph 2, line 12: Skeletal survey was done at the age of 4 months and it showed J-shaped sella turcica and mild beaking of the inferior lumbar vertebrae.

Physical examination was not suggestive of any hepatosplenomegaly. In the absence of any clinical indication, abdominal imaging was not done.

  • Line 92: The author described that “He had 100% engrafting in the myeloid compartment on day 21 post HSCT and remains fully engrafted during the recent clinic visit, one-year post transplant. He continues to be on ERT with laronidase at 0.5 mg/kg every week”. Since HSCT seems successful for this patient, when will the patient plan to stop ERT?

Since multiple studies (Lund et al, 2019 and Guffon et al, 2021) have docmented improved clinical outcome with enzyme augmentation post HSCT, all individuals in our institution receive ERT for 2 years post HSCT. They are evaluated at the 2 year interval regarding the decision on whether ERT needs to be continued beyond 2 years post HSCT.

  • Line 95: What is the data of “N terminal-pro BNP” and “troponin” after ERT and HSCT for this patient? Please add these data in the manuscript.

We have added this information to the manuscript. Please see page 5, paragraph 1, line 17:

His N-terminal BNP continues to be normalized at 1 year post HSCT at 78 pg/mL (reference range: 0-1000) and troponin I at <0.015 ug/L (reference range: 0-0.045).

  • Line 97: In addition to “total antibodies (TAbs) against laronidase”, did the authors also check “Neutralizing antibodies (NAbs) against laronidase” for this patient? If the answer is yes, please also add the data of “Neutralizing antibodies (NAbs) against laronidase” in the manuscript.

Neutralizing antibody has not been checked in this individual.

Round 2

Reviewer 1 Report

Accept

Reviewer 2 Report

The Authors addressed all the reviewer concerns and the quality of the manuscript is now improved.